# Exploring the Impact of a TPSR Program on Transference of Responsibility Goals within a Preschool Setting: An Action Research Study

**DOI:** 10.3390/ijerph17249449

**Published:** 2020-12-17

**Authors:** Fernando Santos, Jacinta Miguel, Paul M. Wright, Cesar Sá, Linda Saraiva

**Affiliations:** 1Polytechnic Institute of Viana do Castelo, School of Higher Education, 901-908 Viana do Castelo, Portugal; joix_lau@hotmail.com (J.M.); cesarsa@ese.ipvc.pt (C.S.); lindasaraiva@ese.ipvc.pt (L.S.); 2Polytechnic Institute of Porto, School of Higher Education, inED Centre for Research and Innovation in Education, 4200-465 Porto, Portugal; 3Department of Kinesiology and Physical Education, Northern Illinois University, 1425 W. Lincoln Hwy, DeKalb, IL 60115, USA; pwright@niu.edu; 4CIEC—Research Centre on Child Studies, University of Minho, 4710-057 Braga, Portugal

**Keywords:** life skills, physical education, responsibility, emotional and social learning

## Abstract

The teaching personal and social responsibility (TPSR) model has been extensively used in a vast array of settings. However, few TPSR studies have focused on preschool settings. The purpose of this action research study was to analyze the experiences of a program leader, her preschool children, and their parents throughout a TPSR program focused on transference of responsibility model goals. The participants were 25 preschool children, six parents, and a program leader involved in a preschool setting located in the north of Portugal. Data were collected through reflexive journaling, participant observations, semi-structured interviews, and focus group interviews. Findings suggest the TPSR model could be a useful instructional model for preschool teachers focused on providing social and emotional learning opportunities to their students. In order to foster transference, parents played a pivotal role in this process and were included in the intervention, which appeared to enhance life skill transfer.

## 1. Introduction

Positive youth development (PYD) has become a commonly used framework in several studies in the area of youth development [1] and within multiple intervention programs [2] to analyze children’s developmental experiences and intentionally foster positive outcomes, such as confidence and respect for others. PYD should be viewed as an asset-based approach that focuses on children’s strengths in contrast with a deficit-based approach that attempts to “fix” challenging behaviors, such as lack of engagement in school or violent behaviors towards others. Physical education (PE) has also been considered a valuable context to foster a vast array of PYD outcomes in multiple countries [1]. Social and emotional learning competencies, including self-awareness, self-management, social awareness, and responsible decision making, have been conceptualized as PYD outcomes [3,4]. Within PE, researchers have highlighted the importance of such competencies [5], and stated these skills may support school readiness and help children thrive in school and other life domains [3]. Preschool children are at a critical development stage for social and emotional learning [6].

An instructional model designed by Hellison [7] has been extensively used to facilitate positive developmental outcomes through sport and physical activity. The teaching personal and social responsibility (TPSR) model focuses on how to intentionally teach personal and social responsibility in, and through, PE [7]. One of the TPSR model’s key pedagogical principles includes the need to create an autonomy-based climate that enables children to become increasingly more responsible for themselves and others. Empowerment guides teachers’ programming as children are encouraged to make decisions and take ownership for their learning experiences. Therefore, TPSR is viewed as much more than a model, and rather, a teaching philosophy that places children’s developmental needs as the primary focus [8]. TPSR involves several responsibility levels to frame developmental goals and teach a range of life skills, such as respect for the rights and feelings of others (level I), effort (level II), self-direction (level III), leadership (IV), and transference of responsibility model goals (level V), based on children’s needs (see Table 1 for an operational definition of each responsibility level). These responsibility levels represent skills to be developed by the preschool teacher which should be defined according to children’s developmental needs. The final objective of TPSR is to foster life skills transfer. In other words, TPSR aims to help young children use the skills learned in the program (e.g., respect, effort) across other life domains.

Researchers have recognized the need for instructional models to be adapted to preschool children’s developmental needs, such as social and emotional learning [9,10]. However, few studies have been conducted to analyze the impact of the TPSR model in preschool settings [9,11]. More research is necessary to explore the adaptations needed to design developmentally appropriate PE programs for preschool children [10]. Typical TPSR-based programs have been conducted in secondary schools with adolescent youth involved in afterschool programs or coaching clubs [12]. Recent research endeavors have started to develop a better understanding of how the TPSR model may be adapted to children’s specific developmental needs [11]. On another note, most of these studies have also focused on the impact of the TPSR model on program leaders’ and students’ perceptions and/or behaviors [13,14]. Fewer TPSR studies have involved the triangulation of perceptions from multiple stakeholders involved in personal and social responsibility development, such as teachers, parents, and children. To date, research endeavors have also mainly focused intentionally on the initial four responsibility levels as transference of model goals has been, in certain cases, addressed unintentionally [12,15]. There is the need to provide insight on how transference of responsibility model goals might be facilitated in preschool settings if the aim is to help children become better prepared to begin formal schooling. Jacobs and Wright [16] proposed a life skills transfer model that reflects the necessary features for life skills transfer to occur. More specifically: (a) program implementation should include a life skill focus so children can internalize and apply life skills, (b) transfer should be viewed as a process that is fostered creating links between the life skills learned in a program and other life domains, and (c) contextual factors such as students’ and teachers’ predispositions and environment (e.g., parents) must be considered.

Transference of responsibility model goals has been studied through self-report data derived from students’ perceptions [17,18]. However, it is relevant to consider that, in preschool education, parental support is particularly important for transference to become a tangible reality. If teacher–parent collaboration exists, students’ learning may be enhanced through having them realize the meaning of a life skill, practicing that life skill, and envisioning transfer of that life skill to other life domains beyond PE. An integrative approach to the framework proposed by Jacobs and Wright [16] leads to an understanding of transference as the association between the nature of a TPSR program, with learning opportunities provided for children by having contextual factors positively influence this process. TPSR-based programs designed for this age range should incorporate parents as partners in the educational process. Researchers studying PYD programs targeting any age group need to further understand how to facilitate transference intentionally and through a clear a set of strategies [7]. An action research design [19,20] enables a thick description of the lived experiences of parents and children involved in the program, as well as the experiences of a program leader while attempting to implement the TPSR model, specifically transference of responsibility model goals. Several action research cycles provided an accurate interpretation of the challenges faced by the program leader, and efforts to develop and implement practices informed by Hellison’s [7] model.

Therefore, the purpose of the present action research study was to analyze the experiences of a program leader, her preschool children, and their parents throughout a TPSR program focused on transference of responsibility model goals within a preschool setting. The following research question drove the present study: How can a program leader foster transference of responsibility model goals within a preschool setting? The present study may add to the TPSR literature by providing insight on how transference of responsibility model goals may occur in preschool settings. Typical TPSR-based programs have been conducted with adolescent underserved youth [21,22]. As such, the responsibility-based strategies used in these programs to foster transference of responsibility model goals should be carefully considered in order to avoid extrapolations to other educational settings, such as preschool. This study may help education professionals understand the challenges faced while fostering TPSR, and identify TPSR strategies for this specific context, as well as add further insight on Pavão et al. [11] study.

## 2. Materials and Methods

### 2.1. The “Early Start to Approaching Responsibility—ESTAR” Program

The TPSR-based program was conducted between September 2017 and December 2018 and followed Hellison’s [7] pedagogical principles and structure proposed for PE sessions. This program was based on previous research conducted by Pavão et al. [11], who created the ‘Early Start To Approaching Responsibility—ESTAR’ program. Expanding on these two components, an empowerment-based climate was promoted by the program leader, as children had the chance to stand in front of class, become a line leader and the leader of the day, as well as help others share materials, and suggest activities, rules, and/or goals. In addition, children were actively involved in the assessment of responsibility behaviors, shared their thoughts with the remaining class, and daily rated their overall performance towards a responsibility goal in a scoreboard placed in the classroom. Responsibility behaviors included respecting others, engaging in activities, and making decisions (see Table 1 for examples of responsibility behaviors). More specifically, after each PE session, children would place their photo on the level of responsibility they thought needed to further develop in future sessions. Children were engaged in this process by suggesting the number of points that should be awarded by the program leader (see Figure 1 and Figure 2 for Hellison’s levels of responsibility and rewards given to children). At the end of the week, a “ceremony” took place to provide feedback on children’s accomplishments. A key difference from traditional TPSR programs was the language used to communicate responsibility goals and desired behaviors. Visual aids and examples were used to reflect the contents of the program in consideration of children’s cognitive and language development. Teacher–student relationships were fostered through individual conversations and frequent interactions with all children. With regards to transference, a “responsibility card” (see Figure 3) was used once a week to communicate with parents, and included a pyramid with Hellison’s responsibility levels, the responsibility objectives developed that week, and two at-home activities (e.g., could you tell your mother/father about the activity you did today in PE?). The final part of the “responsibility card” included a section for parents to provide feedback on how the activities were conducted, and future suggestions. All these adaptations were based on recommendations in the literature [10] and considered children’s needs, cognitive competence, and the Portuguese PE curriculum. To assess model fidelity, several procedures were conducted [7,13]. First, a TPSR implementation checklist was filled out by the program leader after each session, based on TPSR goals. Two sessions were observed by one of the co-authors that also used the TPSR implementation checklist. Second, an expert on the TSPR model, with more than 20 years of experience, served as an external consultant, monitored model implementation, and provided suggestions and adaptations needed to increase model fidelity.

### 2.2. Participants

The participants in this study were 25 preschool children, six parents and a program leader involved in a preschool setting located at north of Portugal. Children were between four and five years old (15 females and 10 males) and had attended this preschool setting for the last two years. Prior to the TPSR-based program designated “Early Start To Approaching Responsibility—ESTAR”, created by Pavão et al. [11], children had not been exposed to any TPSR-based program, and had two one-hour PE sessions per week delivered by a generalist teacher. The program leader was new to the TPSR model, and was initiating her practicum.

### 2.3. Reflexive Journaling and Participant Observations

Prior to data collection, ethical clearance was given by the school’s pedagogical board and the first author’s polytechnic institute to conduct the present. The authors explained the study’s objectives and implications as parents/tutors also consented to participate in the study. Guided by previous research [23], the program leader kept a reflexive journal to document perceived responsibility behaviors presented by the children before the intervention and to identify challenges, define strategies, and clarify the beginning and end of each action research cycle throughout the intervention. The reflexive journal played a pivotal role in enabling the program leader to define challenges within each action research cycle and monitor children’s behavioral change throughout each cycle. Prior to implementing the TPSR model, and during a two-week period, the program leader engaged in four participant observations and reflected on children’s developmental responses to the program that was being developed (i.e., without any TPSR focus), and used this stage to design a set of evidence-based responsibility goals and strategies. Based on these findings and recommendations provided by Casey [19] on action research protocols, the first action research cycle was created, and enabled the program leader to engage in the reflexive process of defining the most beneficial strategies and goals and begin the implementation of the TPSR model. Immediately after each session, the program leader registered her thoughts and interpretations and discussed them with the first author, who provided guidance (i.e., once a week) and also monitored the journal.

### 2.4. Focus Group Interviews

Focus group interviews [24] were used to capture children’s experiences and perceived outcomes at four time points throughout the TPSR-based program. The guide was used at three time points in the program. The first part of the guide included the following questions: (a) What did you do in today’s session?; (b) How did you feel throughout today’s session while working on responsibility?; (c) What responsibility life lesson did you learn today?; and (d) What you would like to improve in the next session? The second part of the guide was conducted with a stimulated recall technique [25] as the interviewer used visual aids (i.e., materials that prompted children to recall previous events) to address responsibility behaviors and activities developed that day to prompt the children on their experiences (e.g., In this activity were you able to respect your classmates?). The focus group interviews were conducted with the preschool children in a developmentally appropriate manner to make sure the children were comfortable and understood the questions posed. This was facilitated by the stimulated recall component. The focus group interviews were conducted on the same day the discussed points included in this section occurred, to avoid problems of recall, and lasted 12 min on average.

### 2.5. Semi-Structured Interviews

Semi-structured interviews were conducted with the parents to understand their perceptions towards the importance of TPSR (e.g., What is the role played by PE in your child’s development?), strategies used outside the school environment (e.g., What strategies do you use to foster respect for others?) and the feasibility of implementing the TPSR model (e.g., Do you believe this model helps you foster transference?). Prior to finalizing the interview guide, a pilot interview was conducted, which resulted in major changes. The participants were made aware this interview was not part of any school assessment procedure and were urged to voice their opinion freely. Interviews lasted, on average, 30 min.

### 2.6. Data Analysis

All the data were transcribed verbatim, read, and reviewed on multiple occasions. A label was given to each data source and data collection instrument (e.g., reflexive journal: RJ; parent: P; child: C; focus group: FG). A six-stage model proposed by Braun and Clarke [26] was used to conduct a thematic analysis. This process was led by the first and second authors, as the remaining co-authors served as “critical friends”. Raw data (e.g., In this activity it was difficult to manage challenging behaviors presented by the children as they were able to understand the activity’s goal) were coded into subthemes (e.g., challenges) that were then grouped into themes (e.g., during the TPSR-based program). Theme refinement enabled the research team to reflect on how to frame and organize emergent themes and subthemes. An inductive analysis [26] was used to create themes and subthemes from the dataset (i.e., data-driven approach). Once the themes and subthemes were deemed representative of the dataset, a report was prepared to present the findings. Guided by recent developments in study quality, a relativist approach [27] was favored in order to select the procedures more appropriate with the nature of this research instead of a set of predetermined analytical steps. All the procedures associated with the intervention, data collection and analysis, and findings were described thoroughly to facilitate transparency. The third author, an established expert on the TPSR model, was an external member of the research team with a particular emphasis on model fidelity, peer debriefing, challenging assumptions, and searching for disconfirming evidence.

## 3. Results

The results were organized into two main themes and 12 subthemes. The theme “Pre-intervention” included three subthemes: the need to foster respect for the rights and feelings of others; no concrete TPSR strategies; few strategies to foster transfer used by parents. The remaining theme “TPSR-based program” included the following subthemes: respect for the rights and feelings of others as a developmental goal; rules and rewards as strategies across the curriculum; participation and effort as developmental goals; concrete opportunities for children to persevere; fostering self-direction and leadership; the need for concrete language and visual aids; strategies to foster transfer; forging teacher-parent collaborations; the “responsibility card” as a key strategy.

Pre-intervention. This theme reflects children’s responsibility behaviors prior to the implementation of the TPSR model, and parents’ and the program leader’s perceptions about the responsibility-based strategies needed to foster transference of responsibility model goals and emergent challenges.

The need to foster respect for the rights and feelings of others. Prior to the intervention, children struggled in respecting the rights and feelings of others, as these type of behaviors were extensively present in other subject areas and class routines outside PE sessions. Hence, children were not able to transfer responsibility behaviors from PE to other settings: “As children are participating in the activities […] I noticed an area was not being used. The preschool teacher mentioned this area was not being utilized […] because children just start destroying the materials” (RJ, September). Additionally, children also struggled sharing materials present in the classroom and respecting class rules: “Every time children are in the [play] areas they are asked to store the materials. However, […] they do not care about helping others” (RJ, October).

No concrete TPSR strategies. Although the program leader was concerned about facilitating responsibility-based outcomes and was familiar with the TPSR model, there were no concrete strategies in place to foster transference of responsibility model goals from PE to other life domains. For example, at this stage, the program leader alluded to the need of implementing the TPSR model with a strong focus on transference: “I have to say that the issue of values needs to be further explored because values are crucial for children’s personal and social development” (RJ, September).

Few strategies to foster transfer used by parents. The parents considered aligning responsibility model goals outside the school environment challenging and named a low set of strategies used with this objective:

“… Sometimes parents do not take responsibility. All these values and skills should be developed by parents […] but work is one of the problems. We are not 100% [available] as we would like because we do not have patience or because we are tired” (P5).

TPSR-based program. This theme included challenges faced throughout the TPSR-based program, strategies used, and outcomes associated to each responsibility level.

Respect for the rights and feelings of others as a developmental goal. Based on the highlighted need of facilitating respect for the rights and feelings of others, the program leader attempted to overcome the main challenges related to the adaptation of the TPSR model to a preschool context: “Respect for others needs to be more explored through PE. Children need clear expectations, positive feedback, and recognition to maintain behaviors throughout time” (RJ, September (second author’s comments)).

Rules and rewards as strategies across the curriculum. The program leader used rewards and extrinsic motivation to foster respect for rights and feelings of others: “Every time each children had a good behavior like paying attention and keeping quiet while someone was explaining an activity […] 10 points were awarded” (RJ, October). For example, the program leader designed activities with clear rules and expectations: “Three balls were given to each child randomly. The children who had the ball had to catch their teammates who did not have it. […] The rules were explained such as do not push your teammates, do not take the ball, and respect everyone” (RJ, October). Children reported that these rules and rewards made them want to maintain a positive responsibility behavior: “When someone touched me with the ball […] I stood still” (C1). “Maria pushed Mário and took the ball from him. I did not do that” (C4) (FG1). The program leader implemented other activities within other subject areas to further enhance responsibility outcomes. For example, the program leader used a group dynamic to discuss positive and negative feelings: “Children were supposed to place pictures [that represented several] situations in a happy and sad heart. These hearts were drawn in cardboard. With this activity, children realized that [these] were mainly good situations and these situations are important to have a happy heart” (RJ, November).

Participation and effort as developmental goals. Participation and effort were developed at this stage by creating concrete opportunities for children to persevere.

Concrete opportunities for children to persevere. All motor tasks created included a “never quit” award as children received points and constant incentives for not quitting, and trying hard: “Today children were involved in different motor tasks and received points for giving their best throughout this PE session. A “never quit” award enabled children to focus on a specific outcome” (RJ, October (second author’s comments)). Children demonstrated awareness of the need for persevering throughout these type of responsibility-based activities: “Yes, I never quit, but I was tired” (C3). “I never quit” (C4). “I even helped Inês” (C5) (FG3). At the end of the session, the program leader used reflexive questions to guide children: “Several guiding questions were asked: “Did you respect your colleague?”, “Did you give up?”, “Did you feel tired?”” (RJ, November). Children progressed rapidly and were able to attain the responsibility objectives associated to level I (i.e., respect for others) and level II (i.e., participation and effort): “Children are able to participate and truly engage in the activities. The challenge is to adapt Hellison’s traditional guidelines to develop self-direction and leadership with this group. What can be done? How? This is the next challenge” (RJ, November (second author’s comments edits)).

Fostering self-direction and leadership. Within level III and level IV, several adaptations were made to use teachable moments to integrate self-direction and leadership. Children had the possibility to make choices and work with others according to their cognitive developmental level and competence: “[We played a game] called “cleaning the house”. The children were divided in four groups. Each group had a home made by a bow and inside other materials. The aim for each group was to take their materials to another bow” (RJ, November) and “The program leader had to organize the most part, but their voice was heard and they had to decide what they liked or not” (RJ, November (second author’s comments)).

The need for concrete language and visual aids. In order to conceptualize these responsibility levels and use more concrete language and effective strategies to explain what it meant to be autonomous and lead others, the program leader introduced a story about family, decisions, and leadership with the aim of clarifying these concepts: “In between the next game, I remembered the theme [we discussed that was] family […] Children will be a character (father, mother, son, and daughter) and their task is to clean the house […] this way children understand that […] we need to help each other” (RJ, December). While showing pictures and materials associated with the activities conducted in PE sessions, children demonstrated how they enjoyed these moments and developed this level of responsibility: “Were you able to this without help?” (program leader). “Me and him won twice!” (C3). “Yes, that is right” (C4). “Did you help others?” (program leader). “We worked together” (C2).

Strategies to foster transfer. Throughout the intervention, transference was addressed intentionally in order to generate tangible outcomes.

Forging teacher–parent collaborations. With the aim of facilitating the development of this responsibility level, parents were involved with the transference of responsibility model goals: “If family does not help, if there is no respect […] if there is no collaboration and empathy within the family […] what can we do?” (P4).

The “responsibility card” as a key strategy. Based on the need to involve parents to facilitate transference, the program leader used a “responsibility card” to share responsibility activities developed in PE and objectives with parents who described positive experiences and outcomes: “[These] type of activities is very important for children to understand there are others [meaning “children”] with needs. She needs to know…” (P2) and “They worked [at home] in organizing lunch and shared tasks” (P3). Both parents and the program leader acknowledged this strategy proved to be useful and recognized transference needs to be intentionally addressed to help children become more responsible in PE and in other life settings: “At home, [we should] work on these values every day […] I have a “green face board”. If he does a task I will give him a “green face”, if he does not, I give him a “red face”. If he gets a lot of “red faces” there is no reward” (P3).

## 4. Discussion

The purpose of this action research study was to analyze the experiences of a program leader, her preschool children, and their parents throughout a TPSR program focused on transference of responsibility model goals within a preschool setting. This study provides novel insights regarding how transference of responsibility model goals may occur within a preschool setting [11,28].

Prior to the intervention, the program leader attempted to comprehend parents’ perceptions, and conducted a series of nonparticipant observation to analyze children’s responsibility behaviors and developmental needs. Hellison’s model holds added value for physical educators and preschool teachers that are aiming to teach personal and social responsibility [7,15]. However, it is of the utmost importance to comprehend children’s needs, class routines, and the nature of the intervention in place in order to avoid the creation of a TPSR model misaligned with a specific educational setting. The pre-intervention stage enabled the program leader to define developmental needs and challenges and define an appropriate plan of action. Previous research [29] has alluded to the importance of assessing responsibility behaviors to further our knowledge of a specific educational context. More research is needed to understand how program leaders adapt the TPSR model to fit a specific educational setting.

Throughout the intervention, several adaptations had to be made to adapt the premises of the TPSR model to fit the nature of a preschool setting. These adaptations have many practical implications for physical educators and teachers involved in a preschool setting. Hellison [7] has conceptualized personal and social responsibility development as an intrinsically motivated endeavor. Although this approach is present in a diversity of TPSR-based programs [30,31], preschool children have quite different cognitive, social, and emotional needs and characteristics and, in this case, an extrinsically motivating climate served the purpose of the initial levels of responsibility, specifically, respect for the rights and feelings of others, and participation and effort. Hence, a reward system was utilized in order to enable children to progress from an extrinsically motivating climate to an intrinsically motivated one, which has been considered a developmentally sound expectation with this target population. In addition, children also reflected on their responsibility behaviors through a “responsibility pyramid” that represented a concrete reference for self-assessment. This approach generated positive responsibility outcomes that may have contributed to how children progressed within the responsibility levels. Nevertheless, physical educators and preschool teachers should interpret these findings carefully and understand that an extrinsic motivating climate has a path to an intrinsic one. If only an extrinsically motivating climate is used, then an empowering climate could be difficult to develop in the long term.

Regarding level III and level IV, the program leader also provided developmentally-sound opportunities for children to understand and develop self-direction and leadership. The language used differed from typical TPSR-based programs, as the program leader used other subject areas to provide insight on what “self-direction” and “leadership” meant, and to better define responsibility objectives. Prior to the PE session, “stories” were used to provide context to the responsibility objectives and motor tasks implemented. Preschool teachers and physical educators should use other subject areas and concrete language that may facilitate understanding of responsibility model goals. Within preschool settings, it is necessary to further investigate if a TPSR mandate used across all subject areas may generate more responsibility outcomes than a unique focus on PE. Previous research [32,33] has conceptualized a series of tasks while promoting level III and level IV, more specifically, assigning group leaders to organize and lead practice. However, in this case, less complex tasks were used. These included assigning a leader in a small group on a specific task, and engaging children in discussions about activities. These strategies might help preschool teachers to embed these responsibility levels in their teaching practice [34].

Despite increasing research on the TPSR model, research on transference of TPSR model goals is still scarce [13]. In the present study, transference was developed through a “responsibility card” and by taking advantage of an existing structure utilized to communicate with others. This provided parents with continued opportunities for engagement throughout the intervention. This “responsibility card” strategy was one of the major adaptations made in the program to enable close and far transfer, increase awareness of the transfer process, and formally assess the transfer process. A recent model, proposed by Jacobs and Wright (2017), also highlighted the need to provide transformative experiences that include experiential value (i.e., understanding the value of life skills), motivational use (i.e., deciding to apply a life skill when it is called for), and expansion of perception (i.e., seeing new opportunities for the transfer of life skills) to facilitate transfer. These opportunities were integrated in the current program with the intention of promoting life skills transfer, which has been highlighted as key in previous research (Pierce, Gould, & Camiré, 2017). It is necessary for physical educators and preschool teachers to consider the aforementioned factors alongside the nature of their TPSR programs, and how student learning is operationalize. Parents should also be considered pivotal in this process and could be included to enhance life skills transfer [16].

In the present study, there are several limitations that should be considered. First, the features of this TPSR-based program should be carefully considered by other researchers, to avoid extrapolations to other cultures/educational systems. Second, the implementation process was conducted throughout a short period of time in which children were exposed to other experiences (e.g., extracurricular activities) outside the program which could have influenced their experiences and parents’ perceptions. Third, the program leader had limited teaching experience in preschool and was involved in a practicum which could have hindered the application of the TPSR model. Finally, the observations were not conducted using a systematic observational tool that could have enhanced rigor and provided a more comprehensive understanding about model fidelity.

## 5. Conclusions

This study offers novel insight related to how the TPSR model may be implemented within preschool settings. The preschool teacher’s experiences implementing TPSR suggest the importance of considering the need to establish mechanisms to foster transference and forge meaningful teacher–parent relationships. Findings also highlight the importance of embedding TPSR across the curriculum and adapting typical TPSR strategies to meet preschool children’s developmental stages and needs. Due to the flexible and dynamic nature of the creation and implementation of any TPSR-based program, researchers and physical educators should consider innovative evidence-based strategies that may help overcome the challenges of facilitating responsibility outcomes within preschool settings. Portraying the lived experiences and perceptions of all actors (i.e., children, parents, program leader) provided a unique outlook on how the TPSR could be implemented in a preschool setting. Therefore, we suggest teacher education programs consider using the TPSR model as a guiding framework for physical educators and preschool teachers looking to teach responsibility.

## Figures and Tables

**Figure 1 ijerph-17-09449-f001:**
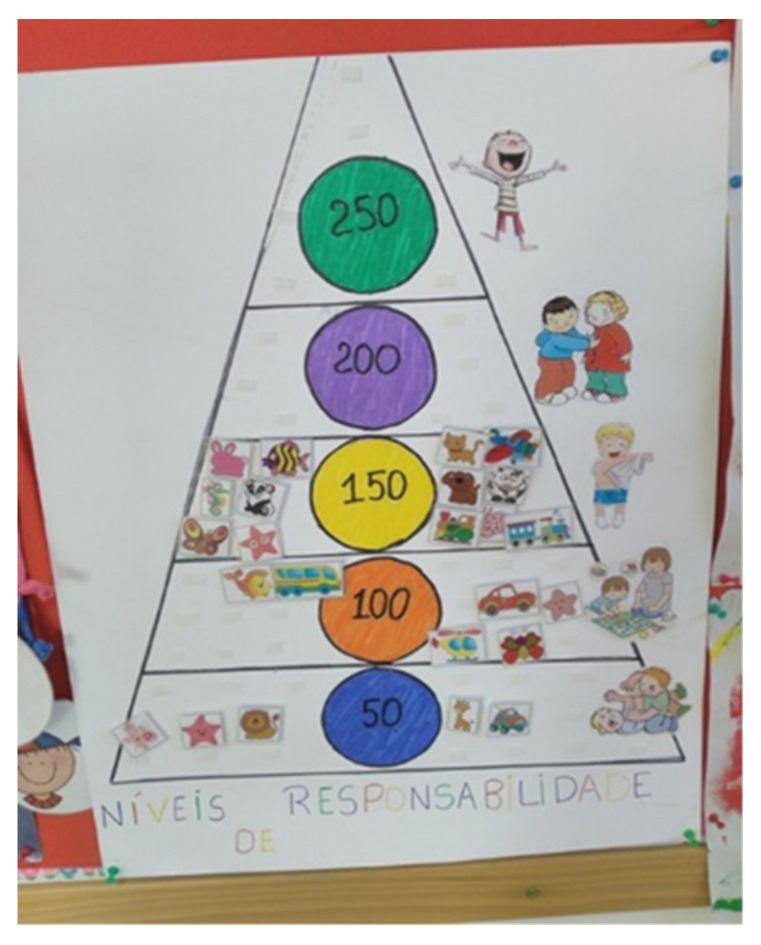
Hellison’s levels of responsibility, with the number of points awarded per level.

**Figure 2 ijerph-17-09449-f002:**
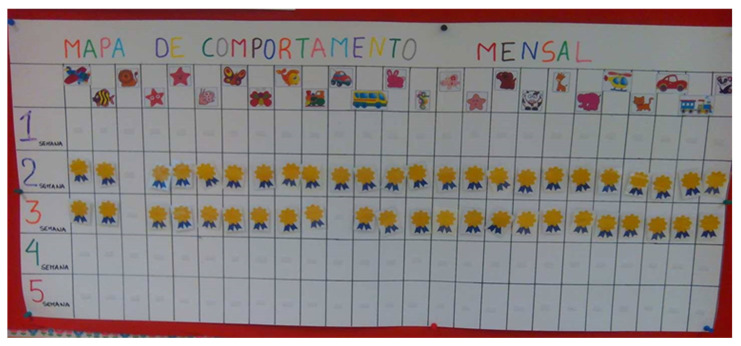
Responsibility board with children’s accomplishments per week represented by a stamp.

**Figure 3 ijerph-17-09449-f003:**
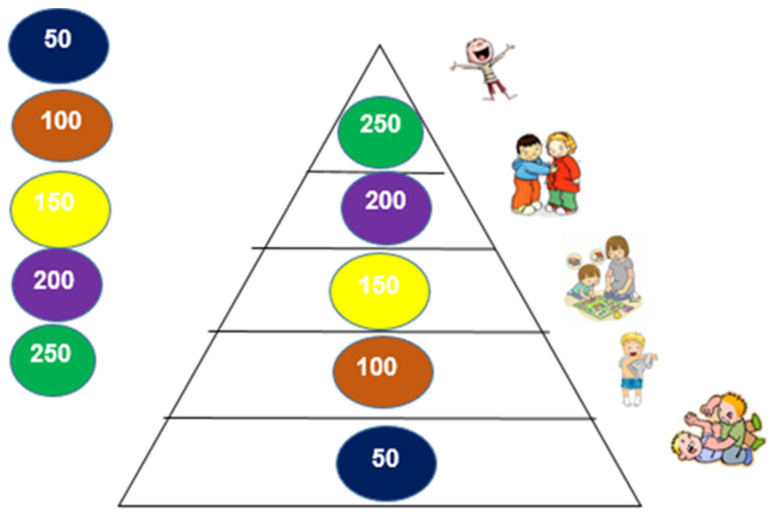
Pyramid sent to parents in the responsibility card.

**Table 1 ijerph-17-09449-t001:** Operational definition of each responsibility level retrieved from https://www.tpsr-alliance.org/tpsr-model/responsibility-levels.

Responsibility Levels	Children’s Responsibility Behaviors
Respect for the rights and feelings of others	Controlling impulses Resolving conflicts peacefully
Participation and effort	Participating in all activities Persisting in difficult tasks
Self-direction	Working independently Setting and working toward goals autonomously Making good choices
Leadership	Helping others Leading others Considering other children’s interests and needs
Transfer	Understanding and applying these skills beyond the preschool setting

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
