# Peer review of "Exploring the Impact of a TPSR Program on Transference of Responsibility Goals within a Preschool Setting: An Action Research Study"

_ijerph, 2020, doi:10.3390/ijerph17249449_

Round 1

Reviewer 1 Report

The study tackles the topic of personal and social responsibility and investigates a TPSR program in a preschool setting. The research presents some potential and interest and investigates a timely topic. However, there are some concerns that the authors should address to revise their manuscript. In the following, I list the major problems and provide indications for revision:

1) There is a lack of explanation of the model adopted. I recommend more elaboration on the TPST model (Hellison, 2011): what psychological and pedagogical principles does the model implement? How are the levels (I-V) conceived? How was the model adapted to preschool children? I also suggest the addition of an explicative figure that gives evidence of the levels, as well as examples of items/constructs for each level.

2) In section 2. Materials and methods I suggest to present the The 'Early Start To Approaching Responsibility - ESTAR' program first, and then the Participants. I also advice to provide examples of "responsibility behaviours" and to clarify the concept of life skills.

3) As for the methods, the authors should clarify the participants in the focus group (children? teachers?) and how many coders were involved in the thematic analysis.

4) The incipt of the Results says that "The results were organized in two main themes and 12 subthemes. However, I did not understand what these themes and subthemes are. Overall, I suggest a revision of the organisation of the results according to themes or any other useful criteria. The current organisation lacks a clear presentation.

5) The discussion and the conclusions suffer from a poor presentation of the model and of its constructs so that the reader does not understand how the multiple level model was effectively implemented.

6) There is an alignment problem with figures 1 and 3.

Author Response

Reviewer 1

The study tackles the topic of personal and social responsibility and investigates a TPSR program in a preschool setting. The research presents some potential and interest and investigates a timely topic. However, there are some concerns that the authors should address to revise their manuscript. In the following, I list the major problems and provide indications for revision:

  • Response: Thank you for your positive comments and all your recommendations.

1) There is a lack of explanation of the model adopted. I recommend more elaboration on the TPST model (Hellison, 2011): what psychological and pedagogical principles does the model implement? How are the levels (I-V) conceived? How was the model adapted to preschool children? I also suggest the addition of an explicative figure that gives evidence of the levels, as well as examples of items/constructs for each level.

  • Response: In light of the reviewer’s comments, we have revised the introduction/literature review and expanded on the TPSR model (see pages 2, 3, 4 and table 1).

2) In section 2. Materials and methods I suggest to present the The 'Early Start To Approaching Responsibility - ESTAR' program first, and then the Participants. I also advice to provide examples of "responsibility behaviours" and to clarify the concept of life skills.

  • Response: We have now included the 'Early Start To Approaching Responsibility - ESTAR' program first, as well as included examples of responsibility behaviors in Table 1. We have also attempted to clarify the concept of life skills in the introduction (page 2 lines 49-57).

3) As for the methods, the authors should clarify the participants in the focus group (children? teachers?) and how many coders were involved in the thematic analysis.

  • Response: We have explained that focus group interviews were conducted with children (see page 7 lines 228-229) and clarified who lead the analysis (see page 7 lines 246-247).

4) The incipt of the Results says that "The results were organized in two main themes and 12 subthemes. However, I did not understand what these themes and subthemes are. Overall, I suggest a revision of the organisation of the results according to themes or any other useful criteria. The current organisation lacks a clear presentation.

  • Response: Thanks for pointing this out. We have reorganized the results to make clear the themes and subthemes that derived from the analysis.

5) The discussion and the conclusions suffer from a poor presentation of the model and of its constructs so that the reader does not understand how the multiple level model was effectively implemented.

  • Response: Based on the changes made to the introduction/literature highlighted above, efforts have been made so the discussion and conclusions may potentially be better interpreted by the reader.

6) There is an alignment problem with figures 1 and 3.

  • Response: We have formatted figures 1 and 3.

Reviewer 2 Report

  1. Research objectives should be listed in a clear and concise manner. One purpose is verbalized, however, several tasks are listed.
  2. if the purpose is to analyze the experience.., in this case in the conclusion section the components of this experience should be clearly listed for the audience to understand concrete results of the research.
  3. Some justification is required why 25 children and 6 parents are ok ad the research population. The technique of the focus group is mentioned, however, some methodological support from academic tradition is necessary 
  4. The results section holds a fully narrative nature. However, at least some bullets would contribute to better perception of the text via its structural visualization. The comments might have been sorted out  according to their types, and not just presented as the narrative flow
  5. Conclusion either needs expansion or can be transformed into concluding remarks, or just added to the discussion section.
  6. Some parts in the results section refer to the procedure lines, FOR INSTANCE 282- 300,264-269

Author Response

Reviewer 2

  1. Research objectives should be listed in a clear and concise manner. One purpose is verbalized, however, several tasks are listed.
  • Response: We have included a research question to complement the purpose of the study. Based on the nature of an action-research study, the purpose and research question should be broad and encompassing in nature (see pages 3 and 4).
  1. if the purpose is to analyze the experience.., in this case in the conclusion section the components of this experience should be clearly listed for the audience to understand concrete results of the research.
  • Response: We have now added a section at the end of the ‘discussion and conclusions’ to better reflect the key findings of this study, specifically the nature of the experiences lived by the program leader (page 12 lines 443-447).
  1. Some justification is required why 25 children and 6 parents are ok ad the research population. The technique of the focus group is mentioned, however, some methodological support from academic tradition is necessary 
  • Response: Thanks for pointing this out. Based nature of an action-research study, participants and data collection methods are determined inductively and considering the diverse action-research cycles. This study involved an intervention towards a specific group of children which drove the sampling procedures. We have added a reference to support the use of focus group interviews (page 7 line 219).
  1. The results section holds a fully narrative nature. However, at least some bullets would contribute to better perception of the text via its structural visualization. The comments might have been sorted out  according to their types, and not just presented as the narrative flow
  • Response: We have reorganized the results to make clear the themes and subthemes that derived from the analysis and facilitate the reader’s analysis.
  1. Conclusion either needs expansion or can be transformed into concluding remarks, or just added to the discussion section.
  • Response: Per the reviewer’s recommendation, we have integrated the conclusions within the discussion.
  1. Some parts in the results section refer to the procedure lines, FOR INSTANCE 282- 300,264-269
  • Response: Thanks for mentioning this. We have kept this information as it helps understand the narrative behind the multiple action-research cycles.

Reviewer 3 Report

Dear authors,

I read your article very carefully. Unfortunately the manuscript presents format and structure issues that can be significantly improved. First of all you need to read the instructions for authors’ (https://www.mdpi.com/journal/ijerph/instructions) and format your manuscript just like the template. Below you will find some major points in the article which needs clarification, refinement, reanalysis, rewrites and/or additional information and suggestions for what could be done to improve it.

Introduction (Section 1)

From the section 1 (introduction) the aim and / or objectives of the study and / or hypotheses and / or research questions are absent or unclear, and which should be numbered and clearly written.

To help you, I quote some questions (as list of points) so that it can be included in your introduction:

-What is the importance of making this study / contribution that it brings to the literature in the field?

-Why should readers be interested?

-What problem/ gap resolve/fill this research?

-To fill this gap (resolve this problem) what solution / intervention / benefits does this research bring? (in other words, how the proposed study will remedy this deficiency/gap/problem and provide a unique contribution to the literature).

-What is the research question which address to the purpose of the research?

Some of these you have already included, however this section should be reviewed and updated.

Materials and Methods (Section 2)

From this section are missing some points and information, e.g., the type of methodology, information about the methods you used, etc.

Also, in your abstract you write that you have done interviews and focus groups, while in your manuscript you write that the interviews have taken place within the focus groups. As methods they are different based on the theory of a research methodology. If this method is new, or there is etc it should be mentioned and documented.

In summary, this also needs a revision and better presentation so that anyone who reads your article can understand it all and not have difficulty reading.

Results (Section 3)

The results as they are presented now are a bit chaotic. This section should also be reviewed and organized to make it easy to read. You could, for example, summarize your results and present them in tables or any other way you think is more correct. If you do them in tables it will be even easier then to connect to the discussion you have in the next section.

Discussion (Section 4)

At several points in this section there are literature review items. You could move this, but in a smaller format, in section 1 (introduction) or delete them. In this section you should comment on your conclusions based on the literature and refer to the results (e.g,in the tables if you present your results in this way).

Conclusions (Section 5)

You will have to write new conclusions, because now they look a little poor.

Other remarks / comments / suggestions:

For your convenience, a new section may be created after the introduction as a bibliographic review. If this is done, then the introduction will only include the purpose, objectives, etc.

You can also merge the discussion and conclusions sections into one section (as discussion and conclusions).

Author Response

Reviewer 3

Dear authors,

I read your article very carefully. Unfortunately the manuscript presents format and structure issues that can be significantly improved. First of all you need to read the instructions for authors’ (https://www.mdpi.com/journal/ijerph/instructions) and format your manuscript just like the template. Below you will find some major points in the article which needs clarification, refinement, reanalysis, rewrites and/or additional information and suggestions for what could be done to improve it.

  • Response: Thank you for all your recommendations.

Introduction (Section 1)

From the section 1 (introduction) the aim and / or objectives of the study and / or hypotheses and / or research questions are absent or unclear, and which should be numbered and clearly written.

To help you, I quote some questions (as list of points) so that it can be included in your introduction:

-What is the importance of making this study / contribution that it brings to the literature in the field?

-Why should readers be interested?

-What problem/ gap resolve/fill this research?

-To fill this gap (resolve this problem) what solution / intervention / benefits does this research bring? (in other words, how the proposed study will remedy this deficiency/gap/problem and provide a unique contribution to the literature).

-What is the research question which address to the purpose of the research?

Some of these you have already included, however this section should be reviewed and updated.

  • Response: We have made multiple changes to the introduction/literature review to better provide a rationale for the study.

Materials and Methods (Section 2)

From this section are missing some points and information, e.g., the type of methodology, information about the methods you used, etc.

  • Response: We have attempted to add more information to the methods section.

Also, in your abstract you write that you have done interviews and focus groups, while in your manuscript you write that the interviews have taken place within the focus groups. As methods they are different based on the theory of a research methodology. If this method is new, or there is etc it should be mentioned and documented.

In summary, this also needs a revision and better presentation so that anyone who reads your article can understand it all and not have difficulty reading.

  • Response: We have created a section titled ‘focus group interviews’ and another designated ‘semi-structured interviews’ to better differentiate the data collection instruments used. We have also made changes throughout the methods section to clarify the steps taken.

Results (Section 3)

The results as they are presented now are a bit chaotic. This section should also be reviewed and organized to make it easy to read. You could, for example, summarize your results and present them in tables or any other way you think is more correct. If you do them in tables it will be even easier then to connect to the discussion you have in the next section.

  • Response: We have reorganized the results to make clear the themes and subthemes that derived from the analysis.

Discussion (Section 4)

At several points in this section there are literature review items. You could move this, but in a smaller format, in section 1 (introduction) or delete them. In this section you should comment on your conclusions based on the literature and refer to the results (e.g., in the tables if you present your results in this way).

  • Response: We have moved some parts of the discussion to the introduction (page 4) and further developed the study’s conclusions. We have integrated the conclusions within the discussion.

Conclusions (Section 5)

You will have to write new conclusions, because now they look a little poor.

  • Response: We have now revised the conclusions (page 12 lines 443-447).

Other remarks / comments / suggestions:

For your convenience, a new section may be created after the introduction as a bibliographic review. If this is done, then the introduction will only include the purpose, objectives, etc.

  • Response: We have revised the introduction but kept the same structure. Based on all your comments, we believe this section has now more flow and better set ups the study.

You can also merge the discussion and conclusions sections into one section (as discussion and conclusions).

  • Response: We have integrated the conclusions within the discussion.

Round 2

Reviewer 1 Report

I'm happy with the revisions the authors made and I feel that the manuscript can be accepted now.

Reviewer 3 Report

Dear authors,
I have read with interest the revised manuscript.
The manuscript has been significantly improved!
Although some parts can be improved, I think you did a good job and the manuscript can be published.
Congratulations for the effort made to improve the work!